# *Camellia oleifera* Shell Biochar as a Robust Adsorbent for Aqueous Mercury Removal

Fenglin Chen [1,†], Nianfang Ma [2,†], Guo Peng [1], Weiting Xu [3], Yanlei Zhang [4], Fei Meng [2], Qinghua Huang [2], Biao Hu [2], Qingfu Wang [2], Xinhong Guo [1,*], Peng Cheng [5,*] and Liqun Jiang [2,*]

[1] College of Biology, Hunan University, Changsha 410082, China
[2] Guangdong Engineering Laboratory of Biomass High-Value Utilization, Institute of Biological and Medical Engineering, Guangdong Academy of Sciences, Guangzhou 510316, China
[3] College of Chemical Engineering, Beijing University of Chemical Technology, Beijing 100029, China
[4] Shaanxi Collaborative Innovation Center of Industrialization of Traditional Chinese Medicine Resources, Shaanxi University of Chinese Medicine, Xianyang 712083, China
[5] Institute of Environmental Research at Greater Bay Area, Key Laboratory for Water Quality and Conservation of the Pearl River Delta, Ministry of Education, Guangzhou University, Guangzhou 510006, China
* Correspondence: gxh@hnu.edu.cn (X.G.); chengpeng@scbg.ac.cn (P.C.); liqun_jiang2508@126.com (L.J.)
† These authors contributed equally to this work.

**Abstract:** *Camellia oleifera* fruit shell (COS) is an agricultural waste product generated in large quantities by the seed oil extraction industry. Due to its hierarchical thickness structure, COS shows huge potential in constructing porous carbon materials after thermal chemical modification. Herein, a series of COS biochars were synthesized by a carbonization-activation process and achieved excellent mercury removal performance in an aqueous environment. High-temperature carbonization was found to facilitate lignin removal and porosity generation, while retaining hydroxyl and carbonyl groups available for mercury adsorption. A volume of micropores of $594 \times 10^{-3}$ cm$^{-3}$/g with average pore diameter of 1.7 nm was achieved in activated COS biochar. At 550 °C, an adsorption capacity of 57.6 mg/g was realized in 1 mg/L Hg$^{2+}$ solution under different pH environments. This work provides an alternative adsorbent for removing hazardous materials using sustainable bioresources.

**Keywords:** *Camellia oleifera* fruit shell; biochar; adsorption; mercury removal

## 1. Introduction

*Camellia oleifera* fruit shell (COS) is an agricultural waste product with large amounts generated by the seed oil extraction industry [1]. Similarly to other biomass resources, the components in COS include cellulose, hemicellulose, lignin, fat, protein, and inorganics [2]. Due to low slenderness, COS is not applicable in paper manufacturing [3]. Traditional disposal methods, such as incineration or landfill, cause a series of environmental deterioration issues. To deal with that, people have started using fermentation technology to make full use of the carbohydrate components in COS, or directly utilize the COS as fertilizer [4,5]. In the point of anatomical structure, COS was found to have a thickened cell wall with modification potential. The thickness of the cell wall from exocarp to endocarp was observed to decrease. The ultrastructure in the hierarchical walls endows COS with conductive, protective, and mechanical properties [6].

In recent years, a variety of value-added products have been developed from COS. Composites and plastic panels are typical examples of reinforced materials [7]. COS has been processed into electrodes with good stability and recyclability in electrochemical devices [8]. COS was also able to be converted to furfural as a significant biomass platform molecule [9]. Due to its hierarchical wall thickness, COS-derived activated carbon has been developed for electric capacitors and CO$_2$ capture [10]. The above outstanding features render the COS as an ideal component in energy and environmental applications.

However, many challenges still confront COS-derived activated carbon materials, such as morphology control and component regulation in terms of their complex biomass composition and structure.

Mercury (Hg) is a highly toxic pollutant found in rivers and seas near factory facilities [11]. It can exist in the forms of inorganic/organic compounds and elemental/metallic mercury [12]. Labile $Hg^{2+}$ will convert to $CH_3Hg$ and then strongly bind to neuron proteins, finally accumulating in brain issues [13]. Therefore, more attention should be paid to mercury removal from contaminated water. A number of processes have been developed for mercury removal, including membrane filtration, coprecipitation, ion exchange, and physical adsorption [14,15]. Among them, adsorption has quick, cheap, and available features that are considered as the most promising methods for mercury removal. Activated carbon has been used as an adsorbent to remove mercury due to its highly ordered porous structure. Large surface areas, interactive functional groups, and processing cost are key issues in the design of activated carbon. Varieties of chemical-modified activated carbon, such as amine-modified carbon, sulfur-impregnated carbon, and phosphorated carbon, have been commercialized [16–18]. However, the synthesis of cost-effective and high-performance porous activated carbon from bioresources remains a huge challenge.

Herein, activated biochar was synthesized from COS and then utilized in aqueous mercury removal. To obtain a porous and ordered structure, natural COS was first carbonized at different temperatures and then treated with $HCl/HNO_3$. The resulting physical properties as well as chemical bond status were comprehensively characterized, and the effect of activation was revealed by comparing carbonized and activated biochars. The porous structure was evaluated. Finally, activated biochars were applied to aqueous mercury. This work provided an archetype using biomass resources as environmental adsorbent by precise morphology control and component regulation in a carbonization process.

## 2. Materials and Methods

### 2.1. Materials

COS feedstock was collected from Hunan province, China. Before use, it was pulverized to 60–80 mesh and then dried at 105 °C to obtain constant weight. All chemicals and reagents used in this study were of analytical grade, including potassium hydroxide (KOH), hydrochloric acid (HCl), and nitric acid ($HNO_3$).

### 2.2. Carbonization and Activation

Carbonization step: Five grams of COS was placed into a carbonization furnace (OTF-1500X, Kejing Co. Ltd., Shenyang, China). $N_2$ was injected 20 min before the temperature rise and start of the carbonization procedure to the target temperatures of 250, 350, 450, 550, 650, 750 and 850 °C at a heating rate of 3 °C/min to obtain carbonization samples (namely bio-char@temperature). The device was purged constantly with nitrogen.

Activation step: The carbonization samples were mixed with KOH as an activator at the ratio of 1:3, and then calcinated with the temperature ramped by 5 °C/min to 750 °C. To obtain activated porous carbon, the resulting black particles were rinsed with HCl (12 wt%) in deionized water and dried at 60 °C. Five milliliters of concentrated $HNO_3$ (68 wt%) was then added to the above carbon materials and then stirred at 60 °C for 3 h [19]. After vacuum filtering and oven drying at 60 °C, acidified porous carbon was obtained. The biochars activated from different temperatures were placed and sealed in vessels (namely a-biochar@temperature), respectively.

### 2.3. Compositional and Structural Characterization

Elemental composition (CHNS/O) was determined by dry combustion-elemental analysis using an elemental analyzer (Vario EL cube, Hanau, Germany). Surface morphology and elemental distribution were obtained by scanning electron microscopy (SEM-EDS) (Hitachi S-4800, Tokyo, Japan). Functional groups were analyzed by Fourier transform infrared spectroscopy (FTIR) in the wavelength range of 750–4000 $cm^{-1}$ by 64 scans at 4 $cm^{-1}$

resolution with an FT-IR spectrometer (BRUKER TENSOR 27, Massachusetts, Germany) equipped with a continuum microscope and ATR objective. Textural parameters were evaluated based on the nitrogen adsorption/desorption isotherm data over the relative pressure range of 0.01 to 1 performed in a surface area analyzer (MicroActive TriStar II Plus 2.03, Shanghai, China). Before nitrogen adsorption, a 140 mg sample of biochar was degassed at 250 °C in a vacuum atmosphere for 2 h. The parameter for monolayer coverage of adsorbed nitrogen obtained from Brunauer-Emmett-Teller (BET) expression was used to calculate the surface area (SABET) over the relative pressure range of 0.01 to 0.3. The quantity of nitrogen adsorbed at saturation (P/P° = 1) was taken as the total pore volume (VT). The average pore radius was calculated from Barrett–Joyner–Halenda expression 2VT/SABET. Micropore surface area and micropore volume were evaluated by the t-plot method, where the micropore surface area and micropore volume were estimated from the slope and intercept of the linear fit in the low P/P° range. Similarly, the slope and intercept of the second linear equation were fitted in the P/P° > 0.4. The BET adsorption method was performed using an ASAP2020 volumetric adsorption analyzer to measure the surface area, pore size, and pore volume of the biochar.

*2.4. Mercury Adsorption Test*

Hg (II) stock solution (1000 mg/L) was prepared by dissolving $Hg(NO_3)_2 \cdot H_2O$ into deionized water. Initial solutions of varying concentrations were obtained by diluting the stock solution with deionized water. Batch adsorption experiments were performed in sealed 50 mL glass vials. Briefly, the reaction was initiated by mixing a certain dose of adsorbent into $Hg^{2+}$ solutions. The vials were then sealed with stirring on a thermostatic shaker at 800 rpm at 25 ± 2 °C for 24 h. After reaching equilibrium, the suspension was centrifuged at 5000 rpm for 10 min, and the supernatant was collected for the remaining mercury concentration analysis. Each of the batch experiments was carried out at 25 °C to examine the effect of solution pH and initial concentration of mercury on mercury removal with the same amount of adsorbent (0.05 g of each). The pH value of the solution was adjusted with the following buffer solutions: $H_2C_2O_4/NaHC_2O_4$ for pH 3.0; $CH_3COONa/CH_3COOH$ for pH 5.0; $Na_2HPO_4$ /$NaH_2PO_4$ for pH 7.0; $Na_2B_4O_7/NaOH$ for pH 9.0; NaOH for pH 11.0. The pH was adjusted before adding the adsorbent.

To optimize the adsorption conditions, the effect of solution pH on the mercury adsorption was investigated from pH 3 to pH 11 at the initial concentration of 1 mg/L. Additionally, the effect of the initial mercury concentration on the adsorption was examined in the range of 1 mg/L to 1000 mg/L without pH adjustment. The adsorption amount was calculated as follows [20]:

$$Q = V(C_0 - C_e)/m \tag{1}$$

where Q is the adsorption amount (mg/g), m is the mass of the sorbent (g), V represents the volume of the solution (L), and $C_0$ and $C_e$ are the concentrations (mg/L) of mercury ions before and after adsorption, respectively.

The effect of reaction time on mercury removal was determined by batch experiments conducted in duplicate. To study the kinetic model of $Hg^{2+}$ adsorption, a batch experiment was performed at 25 °C for 24 h in which the optimum doses of adsorbent and $Hg^{2+}$ solutions were mixed at the optimal pH value. At predetermined time intervals (from 5 min to 24 h), the reaction was artificially terminated in duplicate vials to attain a transient state. Immediately, the solids and liquids to be sampled were separated by centrifugation. The experimental data from this study were used to determine the adsorption isotherm models and kinetic models. The Langmuir and Freundlich models are commonly used to evaluate which one can better fit the experimental data. In this study, the experimental data were fitted into the Langmuir and Freundlich models. The models could be expressed as:

$$Q_e = Q_m C_e/(1/b + C_e) \tag{2}$$

$$\lg Q_e = \lg k_F + (\lg C_e/n_F) \tag{3}$$

where $Q_e$ is the amount of mercury ions adsorbed onto the activated biochars at equilibrium, $Q_m$ is the maximum adsorption amount (mg/g), b is the adsorption equilibrium experimental constant (1/mg), $C_e$ is the mercury concentration (mg/L) in the solution at equilibrium, and b, $k_F$ and $n_F$ are the adsorption equilibrium constants.

The kinetic adsorption equation was calculated as follows:

$$-\ln(1 - F) = kt + c \qquad (4)$$

where t is the adsorption time, k is the adsorption rate constant, c is a constant, and F is the ratio of the adsorption amount at time t ($Q_t$) to that at equilibrium ($Q_e$).

## 3. Results and Discussion

### 3.1. Morphology

The raw *Camellia oleifera* fruit shell (COS) presented large curved sheets with many folds on the rough surface (Figure 1a), where some homogeneous small pores existed, providing a precursor structure for the fabrication of high-surface-area carbonized and activated carbon biochars. During the heating process of carbonization, the initial small pores gradually cracked into long and thin holes, and then narrowed to regular-sized small pores with diameter ranging from approximately 0.2 to 0.5 µm in carbonized COS (biochars). Meanwhile, the carbon sheets fell apart, and carbon block appeared. Interestingly, with the temperature raised to 550 and 650 °C (Figure 1e,f), the obvious regular small porous structure displayed in Figure 1d (450 °C) was absent, replaced by long and thin pores again. Over 750 °C, well-developed porosities could be detected in the carbonized biochars (Figure 1g,h).

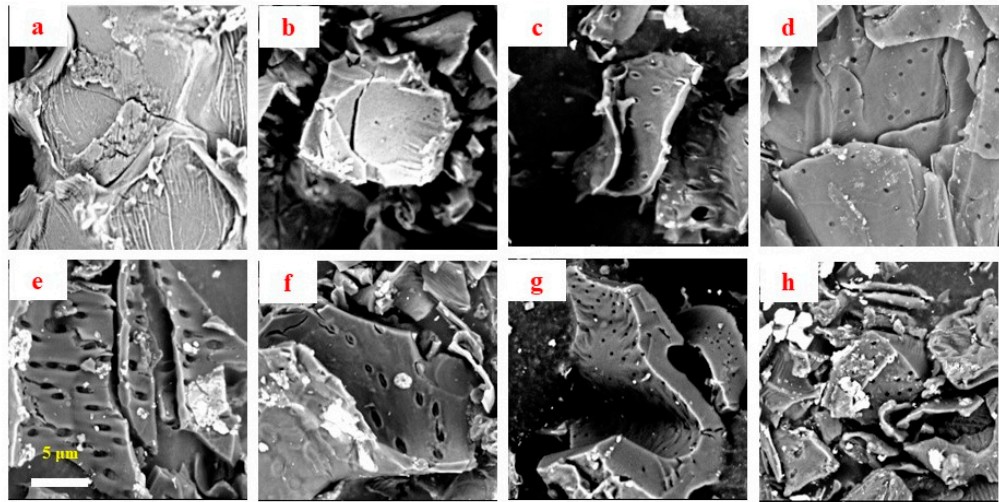

**Figure 1.** SEM images of raw (**a**) and carbonized (**b–h**) *Camellia oleifera* fruit shell (COS) samples obtained at 250, 350, 450, 550, 650, 750 and 850 °C.

The surface morphology of the obtained activated porous carbons (a-biochars) was also examined by SEM, as depicted in Figure 2, which illustrated the porous structure evolution of the biomass after activation. According to previous reports, it was partly caused by a series of reactions with KOH activation, leading to in situ formation of the pores [21]. At a low temperature such as 250 °C, the activation opened the initially existing occlusion holes and exposed the surface of basic microcrystals, thereby facilitating the formation of a large number of extensive homogeneous sponge-like pores at the size of 0.2–1 µm (Figure 2a). With the reaction further proceeding, a large number of micropores and mesopores were found to exist inside the linked macropores, as exhibited in Figure 2b. As the temperature increased, the heat accelerated the impregnation of COS with KOH while promoting the formation of porous structure. The open pores started to expand and

penetrate to more depths. The abundant loose porous structure collapsed, and a regular rich porous structure on the surface was finally generated (Figure 2c,e–g). However, as Figure 2d reveals, under a special activation process, the pores continued to expand and stretch at 550 °C. As shown in Figure 2e,f, with the temperature further rising to 650 and 750 °C, there were pores at the size of about 0.2 μm in a highly ordered arrangement on the surface, which possessed a morphology similar to that achieved with the activation conducted at 450 °C (Figure 2c). However, high temperature (850 °C) resulted in further decreasing the pore diameter to about 0.1 μm.

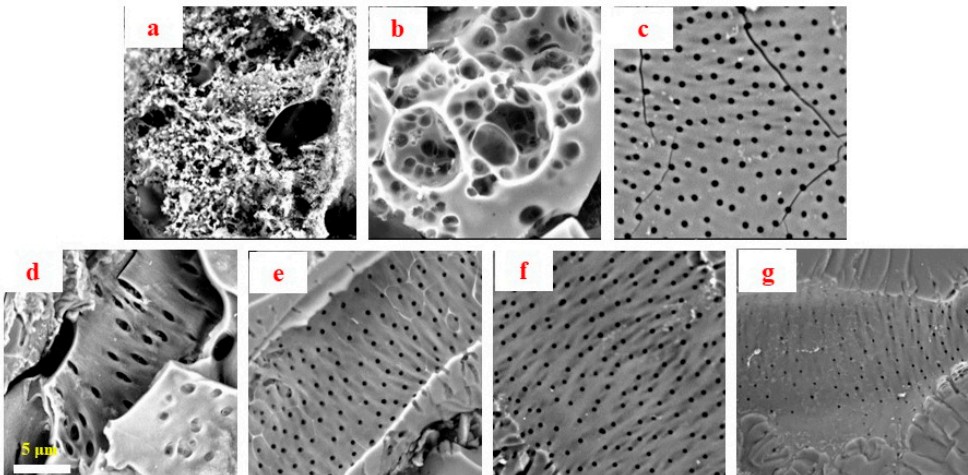

**Figure 2.** SEM images of activated (**a–g**) *Camellia oleifera* fruit shell (COS) samples obtained at 250, 350, 450, 550, 650, 750 and 850 °C.

Compared with raw material, it was found that small pores of similar size replaced the limited original pores on the surface. Comparatively, the porous structure was presented in the two series of samples, while the porous carbons exhibited large differences in morphology. Activated biochars were much richer in porous structure compared to carbonized biochars, and the surface of activated biochars was quite smooth. Based on the above observations, it could be concluded that the activation process played an important role in adjusting the morphology of porous carbons. In addition, both carbonization and activation processes achieved the most porous structure at 450 °C as compared to other heating temperatures.

### 3.2. Component and Functional Groups

Biomass is mainly composed of cellulose, hemicellulose and lignin, but its chemical composition may be different for different biomass types or after carbonization and activation [22]. Table 1 lists the elemental composition of biochars after carbonization and activation. It could be seen that the recovery rate of biochar weight was much smaller at higher temperature. Nearly two times the weight loss was achieved in biochar@850 as compared to biochar@250. This was explained by the high stability of lignin in the biomass feedstock, which required a high temperature (>350 °C) to break down the cross-linking network. At 250 °C, only cellulose and hemicellulose were removed, leading to insufficient pore structure and low performance in adsorption. It was also validated in elemental composition, where the content of C increased while that of H and O decreased, with an elevated C/H ratio. As reported before, this ratio could represent the existence of lignin/carbohydrates in biomass, since the C/H ratio of lignin (11.1) was much higher than that of cellulose (7.5) and hemicellulose (7.2) [23]. With the increase in temperature, the carbohydrates were first removed during carbonization of biochars, and delignification happened at the last stage [24]. Comparing the carbonization and activation processes, it could be found that activation further affected the elemental composition (or lignin-carbohydrate components), where there was only a slight change in the content of N compared to a signif-

icant increase in the content of O. This was proposed to further remove lignin in carbonized biochars after activation, as it could be easily destroyed in the NaOH/acid system. This was also shown in Figures 1 and 2, where the biochar surface became smooth with less physical connection between lignin particles and carbohydrates. No matter which process, the high temperature was more favorable for lignin removal, as reflected by decreased C/H ratios.

**Table 1.** Weight recovery and elemental composition of biochars.

| Sample | Recovery Rate (wt%) | Elemental Content (wt%) | | | |
|---|---|---|---|---|---|
| | | N | C | H | O |
| COS | - | 0.1 | 43.1 | 6.2 | 50.6 |
| biochar@250 | 59.7 | 0.2 | 55.6 | 5.2 | 39.0 |
| biochar@350 | 45.0 | 0.2 | 67.3 | 4.2 | 28.4 |
| biochar@450 | 37.0 | 0.2 | 72.8 | 3.4 | 23.6 |
| biochar@550 | 37.5 | 0.2 | 76.2 | 2.6 | 21.1 |
| biochar@650 | 34.2 | 0.4 | 79.1 | 1.8 | 18.7 |
| biochar@750 | 23.8 | 0.5 | 78.6 | 1.4 | 19.5 |
| biochar@850 | 31.6 | 0.5 | 78.7 | 1.3 | 19.6 |
| a-biochar@250 | - | 0.1 | 11.5 | 1.7 | 86.7 |
| a-biochar@350 | - | 0.2 | 52.2 | 2.7 | 44.8 |
| a-biochar@450 | - | 0.2 | 71.6 | 2.2 | 26.0 |
| a-biochar@550 | - | 0.2 | 70.3 | 2.4 | 27.1 |
| a-biochar@650 | - | 0.2 | 72.5 | 2.3 | 25.0 |
| a-biochar@750 | - | 0.2 | 76.7 | 1.8 | 21.2 |
| a-biochar@850 | - | 0.2 | 73.6 | 2.0 | 24.3 |

The functional groups of carbonized biomass can be analyzed by FTIR spectroscopy. The differences in absorption peaks in different infrared images were related to the activation of carbonized biomass (Figure 3). Both carbonized and activated biochars showed strong absorption peaks at 3000–3750 cm$^{-1}$ and 1640–1675 cm$^{-1}$, representing the stretching vibration of O-H and stretching vibration of C=O. The strong O-H absorption peak was due to the large amount of hydroxyl groups in the structure of three components of biomass (cellulose, hemicellulose and lignin) [25]. The C=O absorption peak of 1640–1675 cm$^{-1}$ was caused by the glucuronic acid carboxyl groups in hemicellulose and the conjugated carbonyl groups in lignin. There was a second shoulder peak beside the C=O absorption peak, which might be caused by adsorbed O-H. The absorption peak at 2820 cm$^{-1}$ of carbonized and activated biochars was caused by C-H stretching vibration, indicating the existence of carbonyl groups, which were also mainly attributed to the three main components (cellulose, hemicellulose and lignin) of lignocellulose. In addition, carbonized and activated biochars showed two small absorption peaks at 1340–1465 cm$^{-1}$. At 1370 cm$^{-1}$ was the bending vibration of C-H, while at 1424 cm$^{-1}$ was the symmetric bending vibration of CH$_2$. The bending vibration of the C-H was caused by the stretching of aliphatic C-H in lignocellulose, while the bending vibration of the latter symmetric CH$_2$ was caused by the carboxyl vibration of glucuronic acid and xylan in cellulose. Different from biochars, a-biochars showed a weak absorption peak at 2260 cm$^{-1}$, which might be due to the stretching vibration of C≡C. It could be speculated that activation only changed the physical structure of the carbonized biomass, generating more pores, but did not change the chemical structure of the carbonized biomass. With the increase in carbonization temperature (250–850 °C), the infrared spectra of biochars and a-biochars had no obvious change, which showed that carbonization temperature had no obvious effect on the chemical structure of biomass.

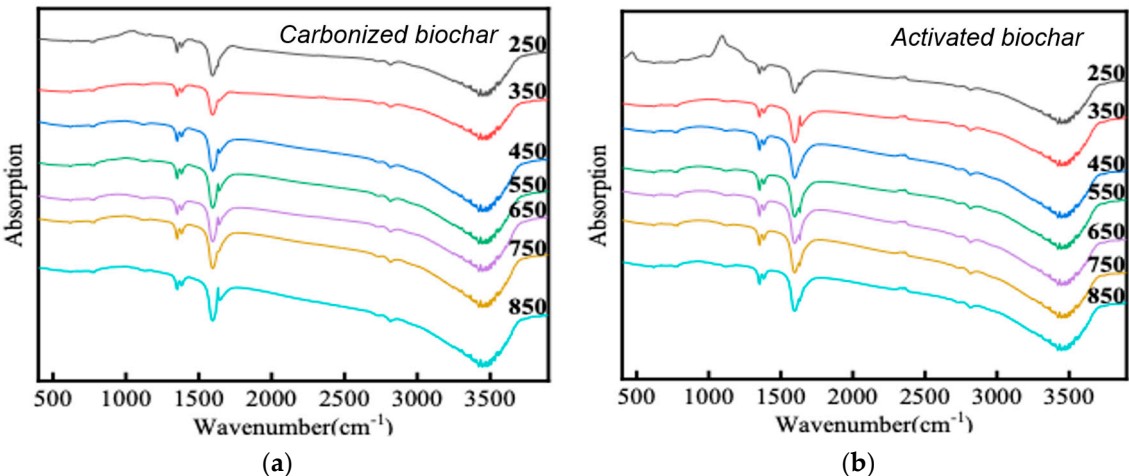

**Figure 3.** FTIR spectrum of carbonized biochars (**a**) and activated biochars (**b**).

### 3.3. Nitrogen Adsorption Capacity

The plots of nitrogen adsorption isotherms and pore size distribution are depicted in Figure 4. The shapes of isotherms suggested that those of biochars prepared at 250 and 450 °C were type II, whereas other isotherms were type I according to the IUPAC classification [26]. Isotherms of biochars prepared at 250 and 450 °C underwent rapid increases in the quantity of adsorption nitrogen at relative pressure <0.1 and >0.9, while a steady process was observed between 0.1 and 0.9. This demonstrated a multilayer adsorption taking place on the external surface of biochars prepared at 250 and 450 °C, with distribution of relatively abundant micropores. Under identical pressure, biochar prepared at 450 °C exhibited much higher adsorption capacity (~4 times) than that of biochar prepared at 250 °C. For other biochars, only a slight increase in the quantity of adsorbed nitrogen was observed at relative pressure of 0.1–0.9, which reflected lower external surface areas of these samples. Figure 4b further validates the micropore structure of prepared biochars, as most of the pores were observed with diameters <2 nm. Only a small amount of mesopores were detected at a relatively narrow pore width range in biochars prepared at 250 and 450 °C.

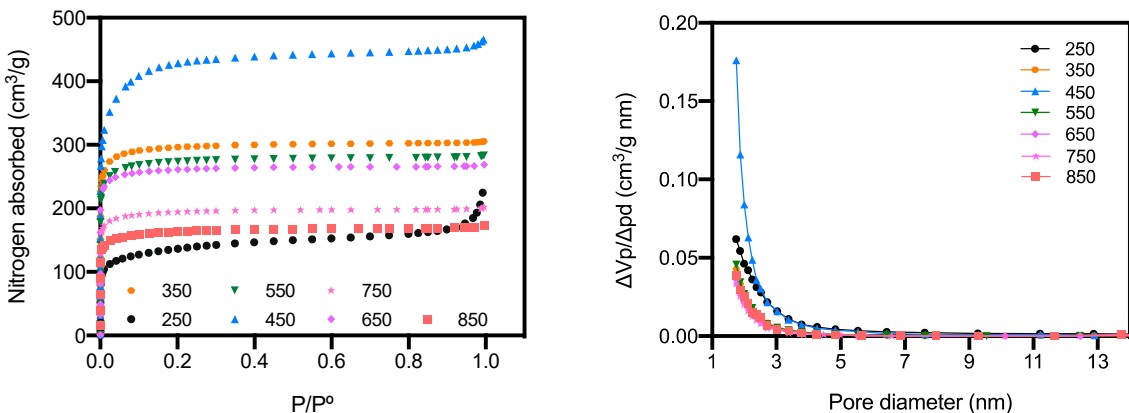

**Figure 4.** Nitrogen adsorption isotherms of activated biochars prepared at different temperatures.

The parameters extracted from nitrogen adsorption isotherms are presented in Table 2. As the heating temperature of biochars increased, both $S_{BET}$ and $S_{Micro}$ increased to reach the maximum of 1614.6 m$^2$/g and 1455.7 m$^2$/g, respectively, at 450 °C, and then slightly decreased. Additionally, micropores dominantly contributed to the total adsorption area (>90%) at temperatures exceeding 250 °C, with the average pore diameters decreasing from 2.8 to 1.6–1.8 nm. Pyrolysis of biomass exceeding 250 °C would produce micropore-rich

biochars with larger surface areas. This could be explained by the removal of volatiles in the biomass and char formation with molecular generation of fused ring with a number > 6 [27]. It was consistent with the compositional analysis discussed above, where hemicellulose decomposition and lignin polymerization took place during pyrolysis [28]. There was a slight decrease in $S_{Micro}/S_{BET}$ at 450 °C, which was probably due to the char over-condensation and pore-widening effect. However, pyrolysis temperatures exceeding 450 °C decreased both $S_{BET}$ and $S_{Micro}$, which demonstrated the destruction of micropore structure derived from thermal instability with carbonization and graphitization occurring. Considering the ionic size of $Hg^{2+}$, micropores could provide suitable trapping channels to the external environment. Therefore, microporous biochars prepared at 450 °C, with $V_{Micro}$ of 0.594 cm$^3$/g and average PD of 1.8 nm, could provide the most abundant storage volumes for $Hg^{2+}$ adsorption in water.

**Table 2.** Nitrogen adsorption isotherm parameters of activated biochars.

| Sample | $S_{BET}$ (m$^2$/g) | $S_{Micro}$ (m$^2$/g) | $S_{Micro}/S_{BET}$ (%) | $V_{Micro}$ ($10^{-3}$ cm$^{-3}$/g) | Avg PD [1] (nm) |
|---|---|---|---|---|---|
| a-biochar@250 | 502.6 | 352.8 | 70.2 | 145.8 | 2.8 |
| a-biochar@350 | 1182.1 | 1124.2 | 95.1 | 432.9 | 1.6 |
| a-biochar@450 | 1614.6 | 1455.7 | 90.2 | 594.0 | 1.8 |
| a-biochar@550 | 1084.8 | 1026.9 | 94.7 | 397.8 | 1.6 |
| a-biochar@650 | 1043.8 | 1004.1 | 96.2 | 386.6 | 1.6 |
| a-biochar@750 | 770.0 | 729.0 | 94.7 | 283.0 | 1.6 |
| a-biochar@850 | 639.0 | 586.5 | 91.8 | 229.8 | 1.7 |

[1] Average pore diameter.

### 3.4. Mercury Removal Performance

Figure 5 shows the mercury removal rate of activated biochar with different adsorbent mass. It was observed that when the adsorbent mass increased from 0.005 g to 0.2 g, the removal rate increased slightly and then reached a plateau at 0.02 g. The mercury removal rate of activated biochar with different adsorbent masses showed that 0.02 g of biochar was sufficient to achieve a high removal rate, as depicted in Figure 5. Therefore, 0.02 g was chosen as the optimal adsorbent mass for further experiments.

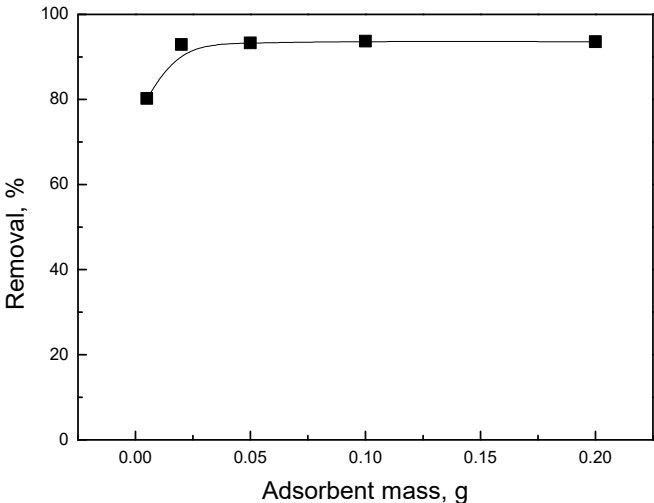

**Figure 5.** Mercury removal rate of activated biochar with different masses.

To optimize the pH for mercury adsorption by the activated biochars, an equilibrium adsorption experiment was conducted with an initial concentration of 1 mg/L over a pH range of 3.0–11.0. Figure 6 illustrates the mercury removal rate under different pH conditions. The results showed that a-biochar@850 had the highest removal rate of $Hg^{2+}$ due to

its highly porous structure and O-H/C=O groups. The high-temperature carbonization process contributed to the activated biochar having a stronger $Hg^{2+}$ removal effect in acidic conditions, which was explained by the accumulated O-H/C=O groups in the biochars. On the other hand, moderate temperature carbonization (i.e., 550 °C) endowed the activated biochar with excellent $Hg^{2+}$ removal performance in basic conditions. This was ascribed to deprotonated O-H groups in a-biochar@550, which further enhanced the adsorption of $Hg^{2+}$ on the biochar pore surface. Biochars processed by low-temperature carbonization had the lowest removal effect, possibly due to insufficient lignin removal and low porosity for $Hg^{2+}$ adsorption.

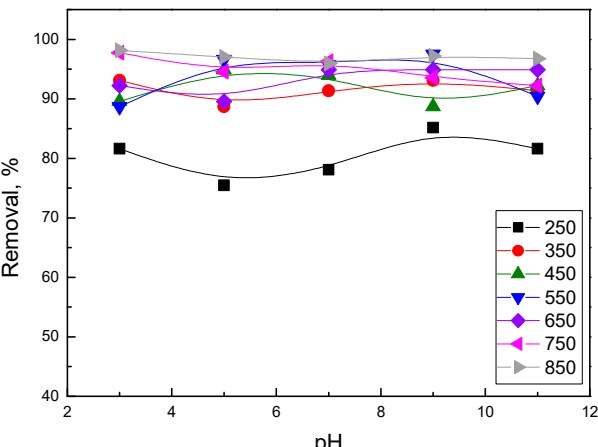

**Figure 6.** Mercury removal rate of activated biochars in different pH environments.

Traditionally, a pH value of 9–10 is considered optimal for $Hg^{2+}$ removal in the $Na_2S$ treatment process. However, in COS biochar-based technology, high-performance $Hg^{2+}$ removal in acid conditions can be achieved for carbonized biochars. Furthermore, while the adsorption amount of activated biochar was not significantly affected by solution pH, it was noteworthy that mercury can be highly adsorbed even at pH 2.

The effect of initial concentration on the removal efficiency for mercury ions is presented in Figure 7. The plot of $C_e/Q_e$ versus Ce for mercury is shown in Figure 8a, while the plot of $lnQ_e$ versus $ln\ C_e$ is demonstrated in Figure 8b. It could be seen that the Langmuir model and the Freundlich model both fitted the data well, which was distinguished from the traditional ion adsorbent [29] and biochar-derived adsorbent [30–32]. From the Langmuir equation, the maximum adsorption capacity of the activated biochars for mercury was calculated to be around 57.6 mg/g. This finding highlighted the high adsorption capacity of the activated biochars for mercury, making them promising adsorbents for environmental remediation.

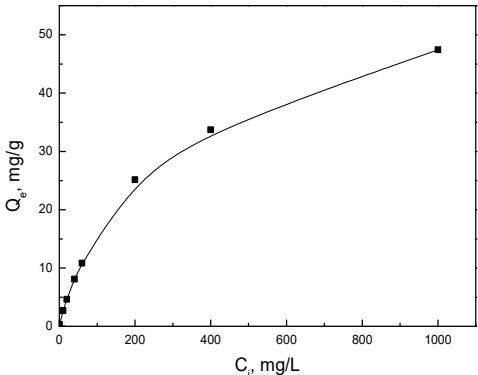

**Figure 7.** Effect of initial concentration on the removal efficiency for mercury ions.

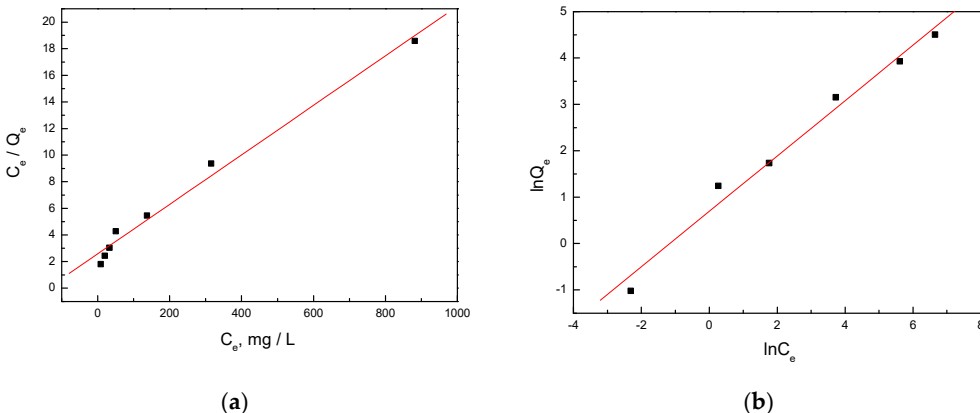

(**a**)　　　　　　　　　　　　　　　　(**b**)

**Figure 8.** Linear fitting using Langmuir (**a**) and Freundlich (**b**) equations for the adsorption of mercury.

The dynamic adsorption results for the activated porous carbon at 550 °C (a-biochar@550) for mercury were presented in Figure 9a, which showed the tendency of adsorption amounts for mercury on the a-biochar in terms of adsorption time. It could be observed that the amount of mercury removed by the adsorbents increased rapidly with the increasing adsorption time and reached adsorption equilibrium after 2 h. The adsorption rate constant for the mercury ions, calculated from the slopes of the plots in Figure 9b, was 0.055 min$^{-1}$. This indicated that the adsorption rates of a-biochar@550 for mercury were rapid, indicating that the biochar had a strong ability to capture mercury.

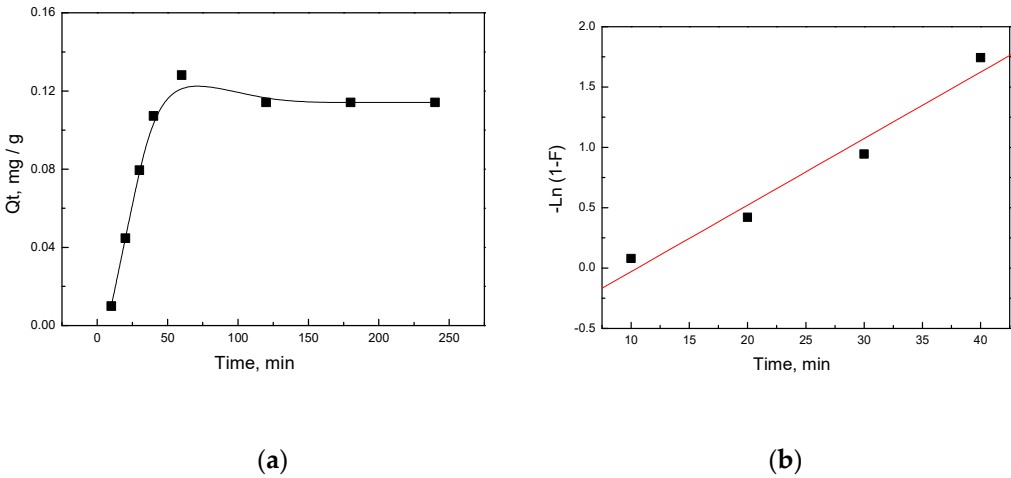

(**a**)　　　　　　　　　　　　　　　　(**b**)

**Figure 9.** Adsorption kinetics of mercury on biochar: (**a**) The dynamic adsorption; (**b**) The kinetic adsorption plots.

## 4. Conclusions

This study synthesized a series of porous biochars by carbonization and activation of COS at a temperature between 250–850 °C, and then applied them to aqueous mercury removal. Due to removal of labile carbohydrates and lignin components, a highly porous structure was formed in the COS biochars. High-temperature carbonization facilitated the production of COS biochars for efficient delignification. The BET test revealed the largest micropore volume of $594 \times 10^{-3}$ cm$^{-3}$/g and an average pore diameter of approximately 1.7 nm in activated COS biochars. Carbonyl and hydroxyl groups were found to be retained during carbonization, which could further contribute to mercury adsorption. As a result, up to 97.2% mercury ions were removed by COS biochar carbonized at 550 °C in 24 h. COS biochars achieved an adsorption capacity of 57.6 mg/g and were proven to be effective in

different pH conditions, thus providing a promising alternative to pollutant adsorbents using biomass resources.

**Author Contributions:** Conceptualization, F.C. and Q.H.; methodology, Y.Z. and Q.W.; validation, Q.H., N.M. and W.X.; formal analysis, W.X.; investigation, B.H.; resources, G.P. and F.C.; data curation, N.M.; writing-original draft preparation, F.C.; writing-review and editing, F.C.; visualization, F.M.; supervision, L.J. and P.C.; project administration, L.J. and X.G.; funding acquisition, L.J. All authors have read and agreed to the published version of the manuscript.

**Funding:** This work was funded by Key Research & Development Project of Hunan Provincial Department of Science and Technology (2019NK2081), GDAS' Project of Science and Technology Development (2022GDASZH-2022010110), the Open Foundation of Shaanxi University of Chinese Medicine State Key Laboratory of Research & Development of Characteristic Qin Medicine Resources (SUCM-QM202206), Guangdong Basic and Applied Basic Research Foundation (No. 2021A1515012063), Jieyang Science and Technology Plan Project (No. 2022DZX027) and Puning Science and Technology Plan Project (No. 202202).

**Institutional Review Board Statement:** Not applicable.

**Informed Consent Statement:** Not applicable.

**Data Availability Statement:** Not applicable.

**Conflicts of Interest:** The authors declare no conflict of interest.

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
