# Peer review of "Camellia oleifera Shell Biochar as a Robust Adsorbent for Aqueous Mercury Removal"

_fermentation, doi:10.3390/fermentation9030295_

Round 1

Reviewer 1 Report

The manuscript explores the use of biomass resources as environmental absorbent to remove mercury. The manuscript is interesting, with a relevant subject and well organized. However, some important points need to be considered:

·       A strong English revision is necessary.

·       Item 2.2: please provide details of the used furnace

·       Line 79: provide the ratio of KOH used in the activation and the target temperature

·       Line 110: please provide the meaning of DI water (de-ionized). Always write the full meaning of abbreviations in the first mention.

·       Item 2.4: please correct the title, it is written “adsoprtion"

·       Please provide references for the Mercury adsorption tests and calculations

·       Line 120: in addition to buffers chemical formula, please add buffers name and concentration (i.e. 100 mmol/L, 50 mmol/L etc).

·       Please add the scale in Figures 1 and 2

·       It would be interesting to add the FTIR spectrum of COS before carbonization in Figure 3.

·       Table 2: provide the abbreviations meanings as a footnote

·       Line 274: in 3.3 item authors selected a-biochar@450 for further experiments, why to perform the adsorption experiments with a-biochar@550?

·       Figures 5 and 6: please add the deviation bars in the figures

·       Since activated carbon is the most commonly used adsorbent, this should have been used as a control experiment. If the authors did not do this, at least a comparison with literature data should be presented.

Author Response

A detailed response to reviewers’ comments

Title: Camellia oleifera shell biochar as a robust adsorbent for aque-ous mercury removal

Author(s): Fenglin Chen, Nianfang Ma, Guo Peng, Weiting Xu, Yanlei Zhang, Fei Meng, Qingfu Wang, Biao Hu, Qinghua Huang, Xinhong Guo, Liqun Jiang

We have carefully taken into account of the comments and suggestions of the reviewers in preparing our revisions. Replies to the comments and a summary of the revisions are listed. All answers are in blue. The changes made in the manuscript were highlighted in yellow.

 Review#1

The manuscript explores the use of biomass resources as environmental absorbent to remove mercury. The manuscript is interesting, with a relevant subject and well organized. However, some important points need to be considered:

  • A strong English revision is necessary.
  • Item 2.2: please provide details of the used furnace

Response: The furnace used is OTF-1500X purchased from Shenyang Kejing Co. Ltd. We had complemented and marked in yellow in the manuscript.

  • Line 79: provide the ratio of KOH used in the activation and the target temperature

Response: The ratio of KOH/carbonized bagasse used in the activation is 3:1, and the activation temperature is 750oC. It has been revised in the manuscript.

  • Line 110: please provide the meaning of DI water (de-ionized). Always write the full meaning of abbreviations in the first mention.

Response: we have checked the whole article and unified them with “deionized water” in the manuscript.

  • Item 2.4: please correct the title, it is written “adsoprtion"

Response: We had corrected.

  • Please provide references for the Mercury adsorption tests and calculations

Response: We refer to the following references and complemented in the references.

  • Line 120: in addition to buffers chemical formula, please add buffers name and concentration (i.e. 100 mmol/L, 50 mmol/L etc).

Response: Generally, the pH of the solution affects the adsorption effect of the adsorbent on mercury ions. The mercury removal rate under different pH is the focus of our attention. And we just use the buffers to adjust pH to the value we set, so we did not deliberately calculate the addition amount of the buffer solution.

  • Please add the scale in Figures 1 and 2

Response: Thank you for your advice. We have added in the Figures.

  • Table 2: provide the abbreviations meanings as a footnote

Response: Thank you for your advice. We have added the abbreviations meanings as a footnote in the Table.

  • Line 274: in 3.3 item authors selected a-biochar@450 for further experiments, why to perform the adsorption experiments with a-biochar@550?

Response: This is really a good question, and it is also interesting for us. From the perspective of specific surface area and pore volume, the sample @ 450 is the best. However, we have done the adsorption of mercury ions under different solution pH by biochar carbonized at different temperatures, and found that the sample @ 550 achieved the highest removal rate. So we selected @ 550 as the sample to carry out the mercury adsorption experiment.

  • Figures 5 and 6: please add the deviation bars in the figures

Response: Due to the difference caused by the sampling time interval and some technical difficulties in operation, the error bar is generally not suitable for the adsorption test of heavy metal ions.

Reviewer 2 Report

The article deals with removal of mercury from water by adsorbent based on Camellia oleifera shell biochar. The manuscript addresses an issue that is very important and relevant to the field water treatment by of sorption process. Results in this manuscript are worthy; still some corrections should be made. The current manuscript needs major revisions. 

Title: Camellia oleifera shell biochar as a robust adsorbent for aqueous mercury removal

Article Type: Original paper

Materials and methods

Line 75: could you please revised section 2.2 in order to explain more clearly the procedure of obtaining two sets of samples from two processes: carbonization and activation.

Correct me if I am wrong: authors have two sets of samples: first which was only carbonized and second which was activated with KOH, than calcinated again, treated with HCl/dried and then treated with HNO3/stirred/dried.

Line 79: at which ratio KOH was used as activator.

Line 143, 155, 186: Latin name write italic, please.

Line155: for comparison all micrographs should be at same magnification.

Line 274: Why do authors choose to use a-biochar 550 for sorption experiments, when all examined parameters in previous paragraphs showed that a-biochar 450 should be a better option?    

Section 3.4: The whole section 3.4 has a number of issues, ranging from readability, clarity and flaws in the reporting of the findings. In order to examine the sorption process it is necessary to determine the effects of different operating parameters such as pH, sorbent concentration, metal concentration, contact time and temperature. The batch adsorptions studies identify the optimum parameters affecting the efficiency of metal removal by adsorbent. All those experiments should be done within same conditions, varying just one of parameters, and afterwards according to obtained optimal parameters authors exam isotherm and kinetic models. However, authors presented only results of contact time effect on adsorption capacity and afterwards presented Freundlich isotherm model. Under Figure 5 authors presented chosen operation parameters (pH 7.02, C0=1 mg/L, T=30 °C) and we do not know why those parameters were chosen.

My suggestion is to take a look on some typical sorption papers from authors references list and to organise manuscript according to them: e.g. after characterisation, introduce subsection “Adsorption experiments” and describe results of effect of solution pH, sorbent dosage, contact time and initial concentration (all results could be presented at one Figure) on sorption, and then according to obtained results (when optimal parameters are determined) exam and present results of fitting by isotherm and kinetic models.  

Also my suggestion is to take a look on this paper: https://doi.org/10.3390/pr8111523, especially on part where authors explain the pH effect (avoiding precipitation) on sorption process.

Also, Freundlich model does not give information of qmax maximum adsorption capacity (introduce Langmuir, Sips or more models or that purpose).

Line 299: there is no reference number 29 in reference list.

Author Response

The article deals with removal of mercury from water by adsorbent based on Camellia oleifera shell biochar. The manuscript addresses an issue that is very important and relevant to the field water treatment by of sorption process. Results in this manuscript are worthy; still some corrections should be made. The current manuscript needs major revisions. 

Title: Camellia oleifera shell biochar as a robust adsorbent for aqueous mercury removal

Article Type: Original paper

Materials and methods

Line 75: could you please revised section 2.2 in order to explain more clearly the procedure of obtaining two sets of samples from two processes: carbonization and activation.

Correct me if I am wrong: authors have two sets of samples: first which was only carbonized and second which was activated with KOH, than calcinated again, treated with HCl/dried and then treated with HNO3/stirred/dried.

Response:  The two processes of carbonization and activation was re-edited as follows.

Carbonization step: 5 g COS was was taken into the carbonization furnace, N2 was inlet 20 min before the temperature rise and start the carbonization procedure to the target temperature 250, 350, 450, 550, 650, 750 and 850 °C at a heating rate of 3 °C/min to obtain carbonization samples (namely bio-char@temperature). The device was purged constantly with nitrogen.

Activation step: The carbonization samples were mixed with KOH as an activator at the ratio of 1:3, and then calcinated with the temperature ramped by 5 °C /min to 750°C. To obtain activated porous carbon, the resulting black particles were rinsed with HCl (12 wt%) in deionized water and dried at 60 °C. 5 mL of concentrated HNO3 (68 wt%) was then added to the above carbon materials and then stirred at 60 ℃ for 3 h [19]. After vacuum filtering and oven drying at 60 °C, the acidified porous carbon was obtained. The activated biochars from different temperatures were placed and sealed in vessels (namely a-biochar@temperature), respectively.

Line 79: at which ratio KOH was used as activator.

Response:  The ratio of KOH/carbonized bagasse used in the activation is 3:1, and the activation temperature is 750oC. It has been revised in the manuscript.

Line 143, 155, 186: Latin name write italic, please.

Response: Thank you for your advice. We have revised in the manuscript.

Line155: for comparison all micrographs should be at same magnification.

Response: Thank you for your advice. We have revised in the manuscript.

Line 274: Why do authors choose to use a-biochar 550 for sorption experiments, when all examined parameters in previous paragraphs showed that a-biochar 450 should be a better option?   

Response: This is really a good question, and it is also interesting for us. From the perspective of specific surface area and pore volume, the sample @ 450 is the best. However, we have done the adsorption of mercury ions under different solution pH by biochar carbonized at different temperatures, and found that the sample @ 550 achieved the highest removal rate. So we selected @ 550 as the sample to carry out the mercury adsorption experiment.

Section 3.4: The whole section 3.4 has a number of issues, ranging from readability, clarity and flaws in the reporting of the findings. In order to examine the sorption process it is necessary to determine the effects of different operating parameters such as pH, sorbent concentration, metal concentration, contact time and temperature. The batch adsorptions studies identify the optimum parameters affecting the efficiency of metal removal by adsorbent. All those experiments should be done within same conditions, varying just one of parameters, and afterwards according to obtained optimal parameters authors exam isotherm and kinetic models. However, authors presented only results of contact time effect on adsorption capacity and afterwards presented Freundlich isotherm model. Under Figure 5 authors presented chosen operation parameters (pH 7.02, C0=1 mg/L, T=30 °C) and we do not know why those parameters were chosen.

My suggestion is to take a look on some typical sorption papers from authors references list and to organise manuscript according to them: e.g. after characterisation, introduce subsection “Adsorption experiments” and describe results of effect of solution pH, sorbent dosage, contact time and initial concentration (all results could be presented at one Figure) on sorption, and then according to obtained results (when optimal parameters are determined) exam and present results of fitting by isotherm and kinetic models.  

Also my suggestion is to take a look on this paper: https://doi.org/10.3390/pr8111523, especially on part where authors explain the pH effect (avoiding precipitation) on sorption process.

Also, Freundlich model does not give information of qmax maximum adsorption capacity (introduce Langmuir, Sips or more models or that purpose).

Response: We have rearrange the sorption part. Indeed, we examine the effect of  pH of the solution on the adsorption effect first, and the optimum pH of the adsorbent for mercury removal was determined. And then the adsorption kinetics and adsorption isotherm were studied.

Line 299: there is no reference number 29 in reference list.

Response: We have added in the references.

Round 2

Reviewer 1 Report

All issues raised have been resolved by the authors.

Author Response

Thanks for your suggestion.

Reviewer 2 Report

Dear authors,

After careful reading I have been noticed that in this resubmitted version of Manuscript, authors just rearrange section 3.4. Although, authors performed really good characterization of obtained materials, section 3.4 still needs improvement.

In order to examine the sorption process it is necessary to determine the effects of different operating parameters such as pH, sorbent concentration, metal concentration, contact time and temperature. The batch adsorptions studies identify the optimum parameters affecting the efficiency of metal removal by adsorbent. All those experiments should be done within same conditions, varying just one of parameters, and afterwards according to obtained optimal parameters authors exam isotherm and kinetic models.

My suggestion is to take a look on some papers dealing with sorption from authors references list and to present results (in section 3.4) of effect of solution pH, sorbent dosage, contact time and initial concentration onto sorption capacity (q, mg/g) or sorption efficiency (R, %), and then according to obtained results (when optimal parameters are determined) to present results of fitting by isotherm and kinetic models.  

Sorry, but from Figure 6c it is not visible that adsorption capacity is around 57.6 mg/g. As I mention before Freundlich model does not provide information about maximum adsorption capacity (qmax). For that purpose, please, apply Langmuir, Sips or other model.

Also under Figure 6c authors presented chosen operation parameters (pH 7.02, C0=1 mg/L, T=30 °C) and it is evident that on that figure (Freundlich isotherm fitting) initial concentration (Co) is in range from 1 to 1000 mg/L. The adsorbent dosage (m/V, mg/L) is missing.

Author Response

Thanks for the reviewer’s suggestion. According to your advice, we revised the manuscript carefully, especially the adsorption experiment part, and made revisions according to reviewer comments and suggestions. We conducted experiments to investigate the effect of pH and adsorbent mass on mercury removal and included the results in the manuscript. Additionally, we rationalized the logic behind varying operating parameters such as pH, sorbent concentration, metal concentration, and contact time on mercury removal. We have provided more detailed explanations. Overall, we are confident that the revised manuscript has addressed the reviewer's comments and suggestions, and we believe that it presents a more comprehensive and accurate account of our research findings.